# Analysis of Changes in Viral Load and Inflammatory Cytokines, as Well as the Occurrence of Secondary Infections, in SFTS Patients Treated with Specific Treatments: A Prospective Multicenter Cohort Study

**DOI:** 10.3390/v16121906

**Published:** 2024-12-11

**Authors:** Jun-Won Seo, You Mi Lee, Sadia Tamanna, Mi-Seon Bang, Choon-Mee Kim, Da Young Kim, Na Ra Yun, Jieun Kim, Sook In Jung, Uh Jin Kim, Seong Eun Kim, Hyun Ah Kim, Eu Suk Kim, Jian Hur, Young Keun Kim, Hye Won Jeong, Jung Yeon Heo, Dong Sik Jung, Hyungdon Lee, Sun Hee Park, Yee Gyung Kwak, Sujin Lee, Seungjin Lim, Dong-Min Kim

**Affiliations:** 1Department of Internal Medicine, College of Medicine, Chosun University, Gwangju 61453, Republic of Korea; kaist-105@daum.net (J.-W.S.); moksha1001@hanmail.net (Y.M.L.); sadiatamanna164@gmail.com (S.T.); ktandms@gmail.com (M.-S.B.); dayz02@hanmail.net (D.Y.K.); shine@chosun.ac.kr (N.R.Y.); 2Premedical Science, College of Medicine, Chosun University, Gwangju 61453, Republic of Korea; choonmeekim@hanmail.net; 3Department of Internal Medicine, College of Medicine, Hanyang University, Seoul 04763, Republic of Korea; quidam76@hanyang.ac.kr; 4Department of Internal Medicine, Chonnam National University Medical School, Gwangju 61469, Republic of Korea; sijung@chonnam.ac.kr (S.I.J.); astralio@naver.com (U.J.K.); favorofgod@hanmail.net (S.E.K.); 5Division of Infectious Disease, Keimyung University Dongsan Medical Center, Daegu 42601, Republic of Korea; hyunah1118@dsmc.or.kr; 6Department of Internal Medicine, Seoul National University Bundang Hospital, Seoul National University College of Medicine, Seongnam 13620, Republic of Korea; yonathan@hanafos.com; 7Department of Internal Medicine, Yeungnam University Medical Center, Daegu 42415, Republic of Korea; sarang7529@hanmail.net; 8Department of Internal Medicine, Wonju College of Medicine, Yonsei University Wonju, Wonju 26426, Republic of Korea; amoxj@yonsei.ac.kr; 9Department of Internal Medicine, College of Medicine, Chungbuk National University, Cheongju 28644, Republic of Korea; hwjeong@chungbuk.ac.kr; 10Department of Infectious Diseases, School of Medicine, Ajou University, Suwon 16499, Republic of Korea; jyheomd@aumc.ac.kr; 11Department of Internal Medicine, College of Medicine, Dong-A University, Busan 49201, Republic of Korea; dsjung@dau.ac.kr; 12Department of Internal Medicine, College of Medicine, Hallym University, Chuncheon 24253, Republic of Korea; easydr@hanmail.net; 13Division of Infectious Diseases, Department of Internal Medicine, College of Medicine, The Catholic University of Korea, Seoul 06591, Republic of Korea; sh-park@knu.ac.kr; 14Department of Internal Medicine, Inje University Ilsan Paik Hospital, Goyang 10380, Republic of Korea; ygkwak@paik.ac.kr; 15Department of Internal Medicine, College of Medicine, Pusan National University, Yangsan 50612, Republic of Korea; beauty192@daum.net (S.L.); babopm@naver.com (S.L.)

**Keywords:** severe fever with thrombocytopenia, viral kinetics, secondary infection

## Abstract

Severe fever with thrombocytopenia syndrome (SFTS) is an acute febrile illness caused by the SFTS virus (SFTSV). We conducted this study to propose a scientific evidence-based treatment that can improve prognosis through changes in viral load and inflammatory cytokines according to the specific treatment of SFTS patients. This prospective and observational study was conducted at 14 tertiary referral hospitals, which are located in SFTS endemic areas in Korea, from 1 May 2018 to 31 October 2020. Patients of any age were eligible for inclusion if they were polymerase chain reaction positive against SFTSV, or showed a four-fold or higher increase in IgG antibody titers between two serum samples collected during the acute and convalescent phases. On the other hand, patients with other tick-borne infections were excluded. In total, 79 patients were included in the study. The viral load of the group treated with steroids was 3.39, 3.21, and 1.36 log_10_ RNA copies/reaction at each week since the onset of symptoms, and the viral load in patients treated with plasma exchange was 4.47, 2.60, and 2.00 log_10_ RNA copies/reaction at each week after symptom onset. The inflammatory cytokines were not reduced effectively by any specific treatment except IVIG for the entire treatment period. Secondary infections according to pathogens revealed four bacterial (26.7%) and one fungal (6.7%) infection in the steroid group. The viral load of SFTSV and inflammatory cytokines cannot be decreased by steroid and plasma exchange treatments. Secondary bacterial infections can occur when steroids are administered for the treatment of SFTS. Therefore, caution should be exercised when choosing treatment strategies for SFTS.

## 1. Introduction

Severe fever with thrombocytopenia syndrome (SFTS) is an acute febrile illness characterized by fever, thrombocytopenia, and leukopenia. It is caused by the SFTS virus (SFTSV), which was first identified in China in 2011 [1]. SFTSV belongs to the *Phlebovirus* genus, a member of the *Bunyaviridae* family [1]. This syndrome is a tick-borne zoonosis, meaning the virus is transmitted to humans through tick bites. The infection spreads from animals to humans and can also be transmitted from human to human [2].

Although it has been 10 years since SFTS was first recognized, we still treat it conservatively due to the lack of a suitable treatment option; as a result, the mortality rate remains high, approximately 6% in China and 20% in Korea [3,4]. Several specific treatments, such as ribavirin, plasma exchange, steroids, tocilizumab, and intravenous immunoglobulin, are administered to patients with severe SFTS. A previous study found that SFTS patients who received plasma exchange treatment early in the onset of symptoms survived for a longer duration than those in the non-plasma exchange group [5]. However, they did not compare cytokine levels between the two groups, and there are limited data on cytokine concentrations in the plasma exchange versus non-plasma exchange groups.

The cytokine storm plays a vital role in the pathogenesis of SFTS. Steroid treatment is considered a therapeutic option to suppress this cytokine storm in severe SFTS patients [6]. Recently, tocilizumab has also been used to mitigate the cytokine storm in SFTS patients. Some studies indicate that tocilizumab has significant clinical advantages when used in combination with corticosteroids [7,8]. Regarding steroid treatment, our previous study and one conducted in Japan found that steroids can increase complications among SFTS patients [9,10]. Additionally, we observed that patients in the steroid group (Acute Physiology and Chronic Health Evaluation II [APACHE II] score < 14) with mild conditions showed significantly shorter survival times compared to those in the non-steroid group [9]. Therefore, therapy associated with cytokine storm modification should be approached with caution. 

We are aware of the risk factors associated with the severity of SFTS in patients. Advanced age is one of these risk factors. In addition, altered mental status, elevated serum lactate dehydrogenase (LDH) and aspartate aminotransferase (AST) levels, creatine kinase (CK) levels, prolonged activated partial thromboplastin time (aPTT), high viral RNA loads in the serum, and the appearance of neurological manifestations, hemorrhagic features, disseminated intravascular coagulation, and multi-organ failure are also observed during severe SFTS infection [11,12,13,14,15]. In particular, changes in viral load after a specific treatment can be a crucial basis for determining the prognosis of SFTS patients, and it is expected that the most suitable treatment for SFTS can be selected based on this. Furthermore, SFTS pathogenesis is associated with a high inflammatory response induced by several inflammatory cytokines, as well as a viral load [16,17]. The cytokine storm is a pathogenic characteristic in fatal SFTS, wherein an exaggerated inflammatory cytokine response leads to an imbalanced immune reaction, causing a systemic inflammatory syndrome that threatens life and results in multi-organ failure [18].

Therefore, we conducted this study to determine the optimum treatment of choice based on the changes in viral load and inflammatory cytokines following four specific treatments. In addition, the occurrence of secondary infections, which can directly affect the prognosis of patients with SFTS, was also evaluated among the several side effects that may occur after the four specific treatments.

## 2. Methods

For this prospective observational study, participants were recruited from 14 tertiary referral hospitals are located in SFTS endemic areas in Korea [19]. Patients of any age were eligible for inclusion if they had a laboratory-confirmed case of SFTS, defined as positive for SFTSV RNA, or a four-fold or higher increase in IgG antibody titers between two serum samples collected during the acute and convalescent phases. 

Viral RNA was extracted from 200 µL of the supernatant from patients’ blood sample using a ZiXpress^®^ Viral Nucleic Acid Extraction Kit in an automated nucleic acid purification system ZiXpress32 (Zinexts Life Science Corp., New Taipei City, Taiwan), and the RNA was eluted using 100 μL of RNase-free water. Complementary DNA (cDNA) was synthesized using SuperScript^®^ VILO™ MasterMix (Invitrogen, Thermo Fisher Scientific, Waltham, MA, USA) at 50 °C for 30 min and 95 °C for 10 min. Real-time reverse transcription polymerase chain reaction for the RNA detection of SFTS virus (SFTSV) was performed in an Exicycler^TM^96 Real-Time Quantitative Thermal Block (Bioneer, Daejeon, Korea), using a pair of primers and probes specific for cDNA and the SFTSV and Lightcycler Taqman Master (Roche, Basel, Switzerland). After synthesizing the following pair of primers and probes that specifically bind to the nucleocapsid protein (NP) gene of the S segment of the SFTSV, real-time reverse transcription polymerase chain reaction (real-time RT-PCR) was used: SQ-F (5′-ACCTCTTTGACCCTGAGTTWGACA-3′), SQ-R (5′-CTGAAGGAGACAGGTGGAGATGA-3′), and SQ-P (5′-[FAM] TGCCTTGACGATCTTA [NFQ-MGB]-3′). After synthesizing the NP gene, a standard curve was drawn using serial dilutions (10^1^–10^8^ copies) of the cloned recombinant plasmid, and the number of viral copies per reaction was determined. The real-time reverse transcription polymerase chain reaction was performed at 95 °C for 5 min, and then 45 cycles at 95 °C for 5 s, and 55 °C for 5 s, before final incubation at 25 °C for 1 min.

To characterize the change of the serum inflammatory cytokines in patients with SFTS treated with specific treatments, the levels of IL-2, IL-6, IL-10, and tumor necrosis factor (TNF-α) were measured in the collected supernatants using human Th1/Th2/Th17 CBA kits (BD Biosciences, Franklin Lakes, NJ, USA) according to the manufacturer’s instructions. Sample acquisitions were performed with a FACS Canto II flow cytometer and analyzed by FCAP Array software version 3.0 (BD Bioscience).

The exclusion criteria were if the patient tested positive for other tick-borne pathogens (like *Orientia tsutsugamushi*, *Anaplasma phagocytophilum*, *Borrelia burgdorferi*, Hantaan or Seoul virus) assessed through PCR assays of acute phase blood samples. This study was approved by the Institutional Review Board of each Hospital. All participants provided written informed consent for the collection and use of their samples for this study.

A healthcare physician interviewed each participant using an epidemiological investigation report to obtain demographic information, medical history, and present illness data containing exposure history. Clinical data, including signs and symptoms of SFTS, laboratory test results, and treatment regimens, were retrieved from patients’ electronic medical records. Secondary infection was confirmed through physical examinations, images such as computed tomography and laboratory tests such as culture and PCR tests using blood, sputum, and urine specimens, performed when patients presented with new symptoms or newly observed signs.

Chi-square or Fisher’s exact tests were used to test for the statistical significance of comparisons between categorical variables, and the Mann–Whitney U-test was used to test for the statistical significance of comparisons between continuous variables. Statistical analyses were performed using SPSS 22.0 for Windows (SPSS, Inc., Chicago, IL, USA).

## 3. Results

A total of 470 patients with suspected SFTS, who were admitted to hospitals and recruited from the southern part of the Korean peninsula, were enrolled between 1 May 2018 and 31 October 2020. Of these, 277 (58.9%) met the eligibility criteria for laboratory-confirmed SFTSV infection. A total of 81 patients underwent additional blood sampling for the analysis of viral copy numbers and cytokine levels at different time points. Among them, 1 patient with scrub typhus and SFTS coinfection, and 2 patients with SFTS-Q fever coinfection, were excluded, leaving 79 patients’ specimens for analysis. The baseline patient characteristics are presented in Table 1.

The median patient age at admission to the hospital was 70 years. Among them, 39 (49.4%) patients were male and 40 (50.6%) were female (Table 1). The median time from symptom onset to hospital admission was 4 days (Interquartile range, IQR, 3–5), and the median duration of hospital stay was 8 days (IQR, 5–11). Here, 15 (18.9%) patients were treated with steroids, 40 (50.6%) patients were treated with immunoglobulin (IVIG), 10 (12.7%) patients were treated with ribavirin, and 14 (17.7%) patients had undergone plasma exchange (PE) (Table 1); 15 patients received more than two specific treatments, and 20 patients did not receive any specific treatment. Of the 79 patients, 23 (29.1%) died. The number of deaths according to each treatment showed that 8 (53.3%) patients received steroids, 12 (30%) received IVIG, 4 (40%) received ribavirin, and 8 (57.1%) underwent PE. Among them, four (26.7%) out of five patients who were treated only with steroids died; 6 (15%) patients died out of 30 who were treated with IVIG; 2 (20%) died out of 5 patients who were treated with ribavirin; and 3 (21.4%) died out of the PE patients (Table 2).

The survival and death analysis of SFTS patients according to treatment showed a statistically significant difference of 53.3% of deaths and 46.7% of survival in the steroid-treated group (with or without other treatment, *p* = 0.022). However, due to the small sample sizes in each treatment group, performing meaningful statistical analysis remains challenging. When steroids were administered alone, 80% of deaths and 20% of survival occurred. The ribavirin treatment group (with or without other treatments) did not show a statistically significant difference with 40% of deaths and 60% of survival (*p* = 0.417), and there was no statistically significant difference in the group treated with ribavirin alone (*p* = 0.580). The IVIG treatment group did not show a statistically significant difference in the 30% of deaths and 70% of survival (*p* = 0.861); with IVIG alone, there was no statistically significant difference (*p* = 0.163). There was a statistically significant difference in terms of 57.1% of deaths and 42.9% of survival in the PE treatment group (*p* = 0.011), and the difference was statistically significant when treated with PE alone too (*p* = 0.038) (Table 2).

The change in the number of viral copies of SFTS between the groups that received and did not receive specific treatment for SFTS was analyzed according to the weeks since the onset of symptoms. The SFTS viral load was compared according to the timing of treatment in the steroid and non-steroid groups to analyze the specific therapeutic effects of SFTS. In the 1st, 2nd, and 3rd weeks of the symptom occurrence, the viral loads of the group treated with steroid treatment were 3.39, 3.21, and 1.36 log_10_ RNA copies/reaction, whereas 3.09, 1.66, and 2.10 log_10_ RNA copies/reaction were found in the non-steroid group. The Mann–Whitney test revealed that the SFTS viral load was significantly higher in the steroid group than in the non-steroid group 2 weeks after symptom onset (Figure 1A). The SFTS viral load was compared according to the timing of treatment in the IVIG and non-IVIG groups. The viral loads of the group treated with IVIG treatment were 3.47, 1.95, and 1.72 log_10_ RNA copies/reaction, whereas 3.0, 1.66, and 2.01 log_10_ RNA copies/reaction were found in the non-steroid group. Patients treated with IVIG did not show a statistically significant difference in terms of viral load compared to patients not treated with IVIG for all periods after symptom onset (Figure 1B). Similarly, patients who received ribavirin also showed no significant decrease in viral load compared to those who did not receive ribavirin treatment (Figure 1C). In the ribavirin group, the viral load changed as follows: 2.24, 3.35, and 2.56 log_10_ RNA copies/reaction each week. The viral loads in patients treated with PE were 4.47, 2.60, and 2.00 log_10_ RNA copies/reaction at 1, 2, and 3 weeks after symptom onset, respectively. These results are similar to those obtained with steroid treatment. In the case of PE treatment, a statistically significantly higher SFTS viral load was found in patients treated with PE than in patients not treated with PE between 1 and 2 weeks since onset of symptoms (Figure 1D).

We evaluated the changes in levels of inflammatory cytokines in patients who received specific treatments for SFTS (Figure 2). We examined the alterations in levels of representative proinflammatory cytokines, namely, IL-2, IL-6, IL-10, and TNF-α, in the pathogenesis of SFTS during hospitalization periods (Hospital Day, HD 0–1, 2–9, 10–19). The mean levels with standard error of the mean (SEM) for each cytokine in patients receiving steroid treatment are as follows: On HD 0–1, HD 2–9, and HD 10–19, IL-2 was 4.34 (0.56), 2.81 (0.19) and 3.10 (0.41), respectively. IL-6 was 35.76 (8.63), 105.71 (60.1), and 90.72 (51.87), IL-10 was 42.12 (10.46), 30.30 (8.42), and 24.55 (1.73), and TNF-α was 45.69 (8.13), 33.96 (5.98), and 37.09 (13.61). In patients who did not received steroids, the mean levels of IL-2 on HD 0–1, HD 2–9, HD 10–19 were 3.68 (0.53), 3.08 (0.34) and 3.05 (0.25). IL-6 levels were 31.51 (16.62), 9.43 (3.68), and 6.27 (1.55), IL-10 levels were 33.73 (3.31), 6.93 (2.53), and 9.44 (5.35), and TNF-α levels were 60.49 (28.84), 15.15 (2.50) and 17.22 (2.69), respectively (Figure 2a). In the steroid-treated group, the representative pro-inflammatory cytokines, IL-6, IL-10, and TNF-α, did not decrease significantly, but rather remained statistically significantly higher by period compared to the non-steroid group.

Among patients who underwent PE, IL-2 was 3.05 (0.22), 4.90 (1.38) and 3.45 (0.58) at HD 0–1, HD 2–9, and HD 10–19. IL-6 was 37.70 (11.31), 535.86 (10.97) and 381.90 (1.80), IL-10 was 29.91 (18.02), 40.60 (16.08) and 35.50 (1.99), and TNF-α was 29.61 (5.82), 141.53 (19.56) and 84.15 (19.73), respectively. In patients who did not undergo PE, IL-2 was 4.61 (0.71), 2.68 (0.09) and 5.46 (2.36) in HD 0–1, HD 2–9 and HD 10–19. IL-6 was 33.15 (12.32), 44.57 (24.44) and 6.78 (1.58), IL-10 was 24.13 (4.50), 6.53 (2.13) and 14.37 (7.13), and TNF-α was 59.11 (20.93), 14.80 (2.10) and 17.32 (2.56) (Figure 2b). The significant elimination of IL-2, IL-6, IL-10, and TNF-α, which we expected from patients applying PE, was not found. Inflammatory cytokines remained at a significantly higher level than those who did not perform PE over the entire period, as with the steroid-treated group.

In Figure 2a, we see that at HD 0–1, HD 2–9, and HD 10–19, in IVIG-treated patients, IL-2 was 4.55 (1.18), 2.88 (0.16) and 3.75 (0.45), respectively. IL-6 was 56.92 (34.02), 39.49 (16.13) and 9.29 (2.44), IL-10 was 30.72 (7.73), 16.07 (6.39) and 5.39 (1.23), and TNF-α was 64.82 (35.44), 21.70 (4.94) and 20.35 (4.20). Patients not treated with IVIG showed IL-2 at levels of 4.31 (0.66), 3.08 (0.39) and 2.75 (0.15) at HD 0–1, HD 2–9 and HD 10–19. IL-6 was 42.92 (18.34), 47.25 (37.71) and 35.70 (7.52), IL-10 was 20.24 (6.15), 7.49 (3.03) and 3.75 (1.02), and TNF-α was 48.19 (18.04), 41.01 (26.02) and 13.75 (1.89), respectively (Figure 2c). Compared to the above-mentioned steroid-treated group and the group given PE, the IVIG-treated group showed significant decreases in IL-6, IL-10, and TNF-α during the entire hospitalization period. Especially, IL-6, a major factor of pathogenesis in fatal SFTS, was found to be statistically significantly reduced even compared to the group not administered with IVIG (*p* = 0.043).

At HD 0–1, HD 2–9, and HD 10–19, in patients treated with ribavirin, IL-2 was 3.19 (0.13), 4.59 (1.44) and 2.80 (0.11), IL-6 was 33.87 (11.04), 152.53 (44.59) and 6.37 (2.02), IL-10 was 16.95 (8.48), 19.84 (11.14) and 3.19 (1.05), and TNF-α was 24.05 (2.66), 23.73 (8.72) and 11.77 (3.12), respectively. In patients not receiving ribavirin, IL-2 was 4.55 (0.73), 2.78 (0.10) and 6.15 (2.91) at HD 0–1, HD 2–9 and HD 10–19. IL-6 was 30.10 (12.36), 51.85 (27.31) and 5.75 (1.66), IL-10 was 27.22 (5.54), 10.58 (3.71) and 9.33 (5.65), and TNF-α was 42.36 (12.26), 18.86 (3.05) and 19.30 (5.21), respectively (Figure 2d). The ribavirin-treated group showed no statistically significant difference between all cytokines we evaluated compared to the group without ribavirin. We evaluated the occurrence of complications, especially secondary infections, between the groups with and without specific treatment. First, secondary infection occurred in 9 out of 79 patients with SFTS. The incidence of secondary infections was compared between the steroid and non-steroid groups. The number of patients with secondary infection was 4 out of 64 patients (6.25%) in the non-steroid group and 5 out of 15 patients (33.3%) in the steroid group, and the incidence of secondary infections between the two groups showed a statistically significant difference (*p* = 0.003). The analysis of secondary infections according to pathogens revealed one bacterial (1.6%), one viral (1.6%), and two fungal (3.1%) infections in the non-steroid group, and four bacterial (26.7%) and one fungal (6.7%) infections in the steroid group (Table 3). In detail, streptococcal bacteremia, hospital-acquired pneumonia (HAP) due to *Pseudomonas aeruginosa* and *Staphylococcus aureus*, urinary tract infection due to *Escherichia coli*, and invasive pulmonary aspergillosis (IPA) due to *Aspergillus fumigatus* were shown in the steroid group. In addition, phlebitis due to *Staphylococcal epidermidis*, influenza, and fungal sinusitis due to *A. fumigatus* were identified in the non-steroid-treated group.

The incidence of secondary infections was compared between the IVIG and non-IVIG groups. The number of patients with secondary infection was 6 out of 39 patients (15.4%) in the non-IVIG group and 3 out of 40 patients (7.5%) in the IVIG group; the incidence of secondary infections between the two groups showed no statistically significant difference (*p* = 0.270). The analysis of secondary infections according to pathogens revealed four bacterial infections (10.3%) and two fungal infections (5.1%) in the non-IVIG group. These included aspiration pneumonia due to *Streptococcus intermedius*, urinary tract infection due to *Klebsiella pneumoniae*, ventilator-associated pneumonia due to *Acinetobacter baumanii,* and HAP due to methicillin-resistant *Staphylococcus aureus* (MRSA). In contrast, one bacterial, one viral, and one fungal (2.5%) infection were found in the IVIG group (Table 3). These include primary bacteremia due to MRSA, influenza, and Candida pneumonia. The incidences of secondary infections in the ribavirin and non-ribavirin groups were also not significantly different between the two groups (*p* = 0.359). The patients with secondary infection numbered 7 out of 69 (10.1%) in the non-ribavirin group and 2 out of 10 (20%) in the ribavirin group. The analysis of secondary infections according to pathogens revealed four bacterial infections (5.8%), including primary *E. coli* bacteremia, HAP due to *S. aureus* and *A. baumanii*, urinary tract infection due to *K. pneumoniae*, one viral infection (1.4%) including respiratory syncytial virus infection, and two fungal infections (2.9%) including candidemia and oral candidiasis due to *C. albicans*, in the non-ribavirin group. The ribavirin group included one (10%) bacterial infection, which is a central line-associated bloodstream infection (CLA-BSI) due to *Staphylococcus epidermidis*, and one fungal (10%) infection, which is IPA due to *A. fumigatus* (Table 3). The incidence of secondary infection was compared between the plasma exchange group and non-PE groups. The number of patients with secondary infection was 7 out of 65 patients (10.8%) in the non-PE group and 2 out of 14 patients (14.3%) in the plasma exchange group; the incidence of secondary infections between the two groups showed no statistically significant difference (*p* = 0.707). The analysis of secondary infections according to pathogens revealed that four bacterial infections (6.2%) included acalculous cholecystitis due to *K. pneumoniae*, urinary tract infection due to *E. coli*, HAP due to MRSA, and *P. aeruginosa* in the non-PE group. In addition, one parainfluenza virus infection (1.5%) and two fungal infections (3.1%), including oral and esophageal candidiasis due to *C. albicans*, were found in the non-PE group. Meanwhile, one bacterial and one fungal (7.1%) infection, which included CLA-BSI due to *S. epidermidis* and IPA due to *A. fumigatus*, were found in the PE group (Table 3). Based on the above results, we can confirm that the risk of secondary infection, especially secondary bacterial infection, was significantly higher in patients treated with steroids than in those not treated with steroids. In the case of other specific treatments, such as ribavirin, IVIG, and PE, there was no statistically significant difference.

## 4. Discussion

Specific treatments, such as ribavirin, PE, steroids, and IVIG, are being presented to severe patients because SFTS leads to a high rate of patients with severe diseases and mortality [1,3,4,5,14]. However, we have not yet secured the best treatment, and we are only applying supportive treatment since we do not have an optimal treatment option yet.

To evaluate the pathophysiology and therapeutic effects of viral infectious diseases, it is essential to study the kinetics of the causative virus. However, the kinetic data of the SFTS virus are still insufficient. Some studies have reported changes in viral kinetics throughout the disease period [20,21]. Shimojima et al. reported that ribavirin did not show any effective reduction in virus load in pre-infected cells, indicating that ribavirin would have no effect on patients who were expected to have very high serum viral loads at the time of hospital admission [22]. Yoo et al. reported that the viral load decreased after treatment in 14 patients who underwent PE alone as a specific treatment for SFTS [23]. However, no studies have prospectively evaluated the changes in viral kinetics and the occurrence of secondary infections before and after the administration of a specific treatment for SFTS.

In this study, data were collected from 277 patients diagnosed with SFTS in Korea from 2013 to 2020, and the therapeutic effects of known specific treatments (steroid, IVIG, ribavirin, and PE) were analyzed in terms of viral load from 79 patients prospectively collected from 2018 to 2020. In addition, we evaluated the presence of secondary infections after administering the aforementioned specific treatments. In this study, patients who received only conservative treatment such as fluids, without steroid, IVIG, ribavirin, or plasmapheresis, were classified into the no specific treatment group. Of the patients enrolled in this study, all patients treated in the ICU were not admitted to the ICU because of the specific treatment strategy used to analyze the treatment effect, and were treated in the ICU from the first day of hospitalization because the severity of the SFTS disease itself was high at the time of admission. Since this study is not a randomized controlled trial, it is likely that primarily mild cases received conservative treatment, which may have led to a 0% mortality rate and a high potential for selection bias. According to our study, among the four known specific treatments, no treatment could significantly reduce mortality. However, in the case of steroids and PE, the mortality rate was significantly higher than that of the patients without these specific treatments. These results can be explained in terms of viral load and secondary infection. We found no significant reduction in the viral load of the SFTSV compared to patients who did not receive any specific treatment. As a result, we evaluated the change in viral load in patients with SFTS treated with steroids, IVIG, ribavirin, and plasmapheresis. Patients with SFTS who received steroids and plasmapheresis had higher viral loads at 2 weeks after symptom onset than those who did not. Based on the results described above, it is difficult to expect a reduction in viral load after the administration of four treatments, each known as a specific treatment for SFTS. In particular, steroids and plasmapheresis have a high risk of increasing the viral load of SFTSV, so we do not recommend these two treatments for SFTS.

Furthermore, we present the results of changes in inflammatory cytokines after specific treatment as a basis for supporting this claim. Cytokine storm can be triggered by infections and is characterized by rapid and prolonged systemic elevation of inflammatory cytokines. Especially, IL-6 and TNF-α are proinflammatory cytokines with central roles in inflammation, and key to cytokine storm-induced mortality [13,16,24,25,26]. On the other hand, serum IL-10 is known as an anti-inflammatory cytokine, and it is a immunoregulatory cytokine in the immune system [27]. However, serum IL-10 is also known to act as an immuno-activated and infectious cytokine in some diseases, including SFTS. In fact, the serum IL-10 concentration was significantly higher in fatal SFTS like IL-6, and can predict poor outcomes in SFTS [16,17,28]. Therefore, the inhibition of and reduction in these inflammatory cytokines is a very important factor in improving the treatment success rate and prognosis of fat SFTS patients. Considering our findings, steroids and PE, which have been known as representative specific treatments, are not very applicable as effective treatments for SFTS. In addition to the aforementioned reduction in viral load, it was found that these two treatment methods were not effective in reducing inflammatory cytokine, as these levels remained higher than in patients who did not receive steroid and PE treatment. Even if our results for IL-10 are analyzed, these conclusions can be interpreted in the same way. Patients with fatal SFTS present with dramatically elevated serum IL-10 concentrations that correlate with disease severity [16,26]. IL-6 and IL-10 are more abundant in patients with severe SFTS, and IL-10 levels increase sharply in acute periods in both fatal and nonfatal cases, while they are low in healthy people or nonfatal cases [16]. These changes in cytokine levels are associated with viral load in the serum and play an important role in fatal SFTS pathogenesis. Kang et al. reported that blocking IL-10 signaling would reduce the production of IL-6, which was found to be high in fatal SFTS patients, and as a result, blocking IL-10 signaling with monoclonal antibodies to IL-10 receptors would be a promising treatment for fatal SFTS [29]. Our study found that steroid, PE, and ribavirin did not have effective IL-10 reduction effects during the entire treatment period, whereas IVIG showed a significant reduction in IL-10 compared to other specific treatments, showing a difference from other specific therapies. Therefore, we expected that IVIG could be used as an effective specific treatment for fatal SFTS patients rather than steroid, PE, and ribavirin.

In addition to our findings, some studies have suggested that steroid administration is inappropriate in patients with SFTS. In 2020, Bae et al. reported that 16 of 45 patients with SFTS (36%) were admitted to the intensive care unit (ICU), of whom 9 (56%) had IPA within eight days, and the mortality rate was reported to be higher in the group in which IPA was confirmed [30]. Except for Bae et al.’s study, there have been few reports of secondary infections by other pathogens, such as bacteria and fungi, in patients with SFTS. In our study, various complications, including an increase in the incidence of secondary infections, were identified, especially in patients treated with SFTS compared to those treated with IVIG, ribavirin, and PE, with a statistically significant difference. This study has confirmed that we should bear the risk of secondary infection, especially bacterial infection, when considering steroids for SFTS treatment. Therefore, our study, which evaluated the change in viral load and the incidence and type of secondary infection after specific treatments for patients with SFTS, can contribute to optimal treatment selection and a reduction in mortality in the future. The limitation of this study is that it was not a randomized controlled trial. Hence, the results of this study have the potential for selection bias, which may affect the relevance of the study. In addition, more severely ill patients were enrolled in this study than patients with mild symptoms, and the majority of the registered patients had to receive specific treatment and ICU treatment, since patients with extreme severity are more likely to receive specific treatments than those with mild to moderate SFTS. As a result, the mortality rate of patients enrolled in this study was higher than the general mortality rate for SFTS. Lastly, another limitation of this study is that only a small number of recruits participated. We speculate that further statistically significant results for the changes in viral kinetics and the occurrence of secondary infections might have been found if more patients had been enrolled, even in treatments other than steroids and PE. Further research on this issue is required, with more patients with SFTS in a prospective cohort to perform the genomic sequences, and to find out if there is any symptom difference among SFTS-positive patients.

In conclusion, the viral load of SFTSV and mortality of SFTS can be increased by steroid and PE treatments. In the case of the PE group, the fact that there was a difference in viral load from the beginning of treatment within 1 week of symptom onset suggests that PE treatment was applied to severely ill patients, which means that the possibility of selection bias cannot be excluded. However, there was no statistically significant difference in viral load within 1 week after symptom onset in the steroid-administered group; the viral load increased more within 2 weeks of symptom onset. In terms of the reduction effects of inflammatory cytokines that play a major pathophysiological role in fatal SFTS, the effects of steroid and PE are difficult to properly predict. On the other hand, IVIG can be predicted to cause an effective reduction in inflammatory cytokines. In addition, secondary infections, especially bacterial infections, can occur when steroids are administered to treat SFTS. Therefore, we should be very careful when choosing a treatment for SFTS, and more reliable studies with more patients are needed to determine an effective treatment for SFTS.

## Figures and Tables

**Figure 1 viruses-16-01906-f001:**
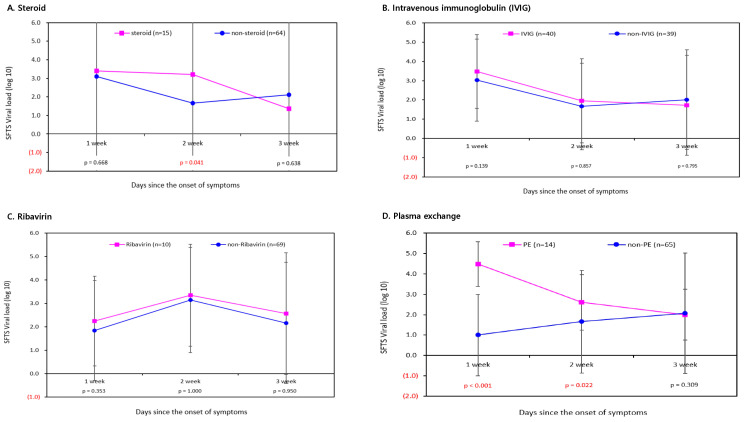
Changes in SFTS viral load according to four specific treatments.

**Figure 2 viruses-16-01906-f002:**
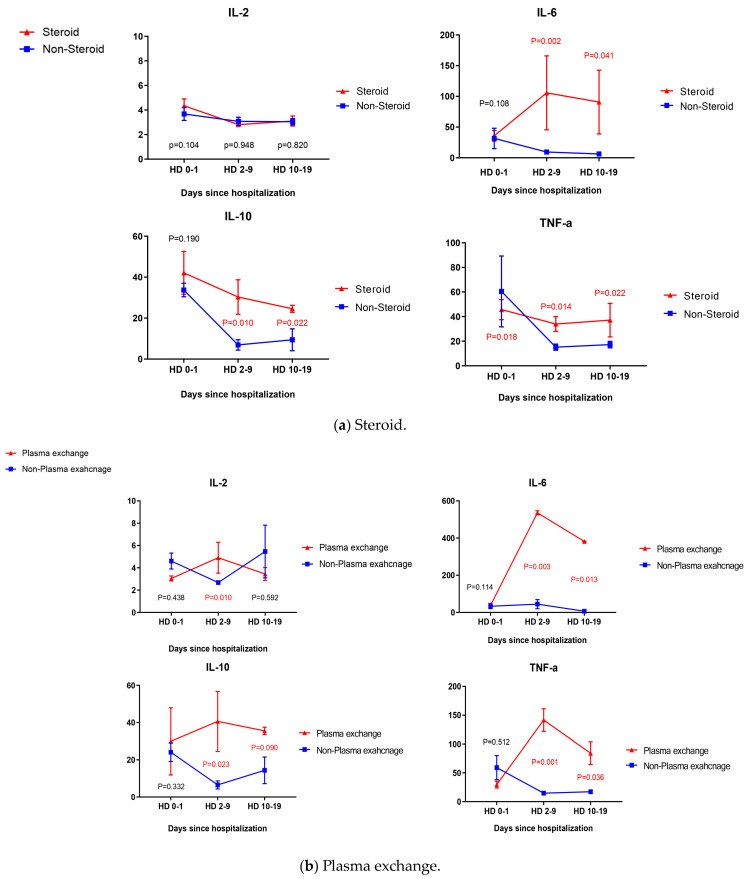
Changes in levels of inflammatory cytokines over time after four specific treatments for SFTS.

**Table 1 viruses-16-01906-t001:** Baseline demographic and clinical characteristics of patients with laboratory-confirmed SFTS across the study period.

Characteristic	Total Patients	Specific Treatments				No Specific Treatment
		Steroid	IVIG	Ribavirn	Plasma Exchange	
Median age, years (IQR)	70 (61–77)	76 (71–83)	71 (61–77)	71 (56–77)	75 (72–81)	66 (55–76)
Patient number (%)		15 (19.0%)	40 (50.6%)	10 (12.7%)	14 (17.7%)	20 (25.3%)
Male	39 (49.4)	5 (6.3)	20 (25.3)	6 (7.6)	5 (6.3)	11 (13.9%)
Female	40 (50.6)	10 (12.7)	20 (25.3)	4 (5.1)	9 (11.4)	9 (11.4%)
Median APACHE II score (IQR)	10 (9–13)	11 (9–15)	11 (8–16)	12 (10–25)	13 (10–17)	9 (6–12)
Median duration of ICU hospitalization, day (IQR)	6 (3–7)	6 (3–10)	6 (2–8)	7 (4–9)	5 (2–6)	0
Mortality, number (%)	23 (29.1)	8 (53.3)	12 (30)	4 (40)	8 (57.1)	0 (0%)

Abbreviations: IQR, interquartile range; APACHE, Acute Physiology and Chronic Health Evaluation; ICU, intensive care unit; IVIG, intravenous immunoglobulin.

**Table 2 viruses-16-01906-t002:** Survival and mortality analysis according to treatment in SFTS patients.

Treatment	Patient Number (n, %)						
	Total (79)		Survival (56)		Mortality (23)		*p*
Steroid	15	19.0%	7	46.7%	8	53.3%	0.022
only Steroid	5	6.3%	1	20%	4	80%	0.01
Ribavirin	10	12.7%	6	60%	4	40%	0.417
only Ribavirin	5	6.3%	3	60%	2	40%	0.580
IVIG	40	50.6%	28	70%	12	30%	0.861
only IVIG	30	38.0%	24	80%	6	20%	0.163
Plasma exchange	14	17.7%	6	42.9%	8	57.1%	0.011
only plasma exchange	4	5.1%	1	25%	3	75%	0.038
No specific treatment	20	25.3%	20	100%	0	0%	NA

Abbreviations: Intravenous immunoglobulin, IVIG; not applicable, NA.

**Table 3 viruses-16-01906-t003:** The incidence of secondary infection according to specific treatments.

	Specific Treatments											
	Steroid(n = 15)	Non-Steroid(n = 64)	*p* Value	IVIG (n = 40)	Non-IVIG (n = 39)	*p* Value	Ribavirin (n = 10)	Non-Ribavirin (n = 69)	*p* Value	Plasma Exchange (n = 14)	Non-Plasma Exchange (n = 65)	*p* Value
Secondary Infection, Number (%)	5 (33.3%)	4 (6.25%)	0.003	3 (7.5%)	6 (15.4%)	0.270	2 (20%)	7 (10.1%)	0.359	2 (14.3%)	7 (10.8%)	0.707
Pathogen												
Bacteria	4 (26.7%)	1 (1.6%)	<0.001	1 (2.5%)	4 (10.3%)	0.157	1 (10%)	4 (5.8%)	0.610	1 (7.1%)	4 (6.2%)	0.890
Virus	0 (0%)	1 (1.6%)	0.626	1 (2.5%)	0 (0%)	0.320	0 (0%)	1 (1.4%)	0.702	0 (0%)	1 (1.5%)	0.640
Fungus	1 (6.7%)	2 (3.1%)	0.518	1 (2.5%)	2 (5.1%)	0.541	1 (10%)	2 (2.9%)	0.272	1 (7.1%)	2 (3.1%)	0.470

Abbreviations: Intravenous immunoglobulin, IVIG.

## Data Availability

Dong-Min Kim had full access to all data in the study and take responsibility for the integrity of the data and the accuracy of the data analysis. All data are available from the corresponding author upon reasonable request.

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
