# Peer review of "Analysis of Changes in Viral Load and Inflammatory Cytokines, as Well as the Occurrence of Secondary Infections, in SFTS Patients Treated with Specific Treatments: A Prospective Multicenter Cohort Study"

_viruses, 2024, doi:10.3390/v16121906_

Round 1

Reviewer 1 Report

Comments and Suggestions for Authors

Authors have compiled clinical results from 79 SFTS patients in Korea. I appreciate how they recognize the critical importance of correct treatment of the patients. There are two main issues that need to be corrected in the manuscript.  One, for the sake of clarity for the reader much of the information can be consolidated in tables rather than writing the results in text. Second, it is unclear how SFTSV were analyzed from patients. It appears that secondary infections are analyzed by PCR from blood samples, however, SFTSV QRTPCRs are done from a cell culture or something else?  This part is quite unclear.  I also have line by line comments for the rest of the manuscript.

Line 36: This should say that patients with other tick-born infections were excluded (Line108-110)

Lines 39-41:  Can these data be put in a graph?  Are the virus copies per mL of blood?

Line 55: Is there a reason why mortality rate is so different in China vs. Korea?

Lines 60-65: Advanced age is the only risk factor and rest of the listed are symptoms of the disease.

Line 84: Was the viral RNA extracted from blood? Were blood cells cultured? How can this be quantitative measure of virus copy in the patient?  I am confused about this.

Line 105: What are these minor modifications?

Line 124: What is the number of recruited patients? How were they recruited?

Lines 125-126: If 277 patients were confirmed to have SFTSV infection from 470 patients, how did the authors arrive at 79 patients in the end?

Line 129:  Who are the “No specific treatment” -group? Why is the mortality 0% in this group?

Line 145: About the treatment group.  What other treatments are for the categories of Steroid, Ribavirin, IVIG, and Plasma exchange?

Lines 147-158:  Since the numbers per treatment group as so small, it is difficult to consider statistical analysis relevant.

Line 209: Please, put the figure headings above the figures.

Line 333: methods were not effective.

Line 353: What is IPA?

Line 354: Except for Bae et al. study (remove apostrophe)

Comments on the Quality of English Language

Please, include much of the data into tables instead of text.

Author Response

Reviewer 1

Comments and Suggestions for Authors

Authors have compiled clinical results from 79 SFTS patients in Korea. I appreciate how they recognize the critical importance of correct treatment of the patients. There are two main issues that need to be corrected in the manuscript.  One, for the sake of clarity for the reader much of the information can be consolidated in tables rather than writing the results in text. Second, it is unclear how SFTSV were analyzed from patients. It appears that secondary infections are analyzed by PCR from blood samples, however, SFTSV QRTPCRs are done from a cell culture or something else?  This part is quite unclear.  I also have line by line comments for the rest of the manuscript.

Response: Thank you for your valuable comment. Almost all the results are included in Table 1, Table 2, and Figures 1 and 2. However, the technical content appears poorly organized, which may reduce readability. For clarity, I have referenced which figures or tables the information pertains to in the results description. I hope this helps readers understand the information better.

“The median patient age at admission to the hospital was 70 years. Among them, 39 (49.4%) patients were male and 40 (50.6%) were female (Table 1). The median time from symptom onset to hospital admission was 4 days (Interquartile range, IQR, 3–5), and the median duration of hospital stay was 8 days (IQR, 5–11). 15 (18.9%) patients were treated with steroids, 40 (50.6%) patients were treated with immunoglobulin (IVIG), 10 (12.7%) patients were treated with ribavirin, and 14 (17.7%) patients had undergone plasma ex-change (PE) (Table 1).”

In your second comment, the line was mistakenly written wrong in our manuscript. We apologize for that. First of all, secondary infection was confirmed through physical examinations, image such as computed tomography and laboratory test such as culture and PCR tests using blood, sputum, and urine specimens, performed when patients presented with new symptoms or newly observed signs. We only collected blood samples from patients and performed PCR with extracted RNA from whole blood of patients. We did not perform any cell culture for this study. We have resided our manuscript as below:

“Viral RNA was extracted from 200 µL of the supernatant from patients’ blood sample using a viral DNA/RNA extraction kit (Cat No. ZP02201) in an automated nucleic acid purification system (ZiXpress-32), and the RNA was eluted using 100 μl of RNase-free water. Complementary DNA (cDNA) was synthesized using SuperScript® VILO™ Mas-terMix (catalog number 11755; Invitrogen, Thermo Fisher Scientific, USA) at 50 °C for 30 min and 95 °C for 10 min. Real-time reverse transcription polymerase chain reaction for RNA detection of SFTS virus (SFTSV) was performed in an ExicyclerTM96 Real-Time Quantitative Thermal Block (Bioneer, Daejeon, Korea), using a pair of primers and probes specific for cDNA and the SFTSV and Roche Master Mix (Lightcycler Taqman Master).”

Further comments are below:

  1. Line 36: This should say that patients with other tick-born infections were excluded (Line108-110)

Response: Thank you for your valuable comment. We have revised our manuscript as per your suggestion and rewrote the line as below:

“Patients of any age were eligible for inclusion if they were only positive against SFTSV, or a four-fold or higher increase in IgG antibody titers between two serum samples collected during the acute and convalescent phases. On the other hand, patients with other tick-born infections were excluded.

  1. Lines 39-41:  Can these data be put in a graph?  Are the virus copies per mL of blood?

Response: Thank you for your valuable comment. We have already input these information into the Figure 1 (a) and (d).

Virus copies are for per reaction of each experiment. Therefore, we have corrected is one as genome copies per reaction.

However, for the better understanding, we have revised the y axis title as “log10 RNA copies/reaction (SFTS)” instead of “SFTS Viral load (log10)”

  1. Line 55: Is there a reason why mortality rate is so different in China vs. Korea?

Response: Thank you for your comment. The main reason appears to be the SFTSV strain. There are six genotypes of SFTSV, and all are present in both China and South Korea. However, the distribution of each genotype varies between the two countries. Previous reports indicate that genotype A, the dominant strain in China, has the lowest mortality rate. In contrast, genotype B-2, which is most prevalent in South Korea, is associated with the highest fatality rate. This accounts for the differing mortality rates between China and South Korea.

Reference:

  • Han, Xiao-Hu, Yue Ma, Hong-Yan Liu, Dan Li, Yan Wang, Feng-Hua Jiang, Qing-Tian Gao et al. "Identification of severe fever with thrombocytopenia syndrome virus genotypes in patients and ticks in Liaoning Province, China." Parasites & Vectors 15, no. 1 (2022): 120.
  • Yun, Seok-Min, Su-Jin Park, Young-Il Kim, Sun-Whan Park, Min-Ah Yu, Hyeok-Il Kwon, Eun-Ha Kim et al. "Genetic and pathogenic diversity of severe fever with thrombocytopenia syndrome virus (SFTSV) in South Korea." JCI insight 5, no. 2 (2020).
  1. Lines 60-65: Advanced age is the only risk factor and rest of the listed are symptoms of the disease.

Response: Thank you for your valuable comment. We have revised our manuscript as per your suggestion and rewrote the line as below:

“We are aware of the risk factors associated with the severity of SFTS in patients. Advanced age is one of these risk factors. , In addition, altered mental status, elevated serum lactate dehydrogenase (LDH) and aspartate aminotransferase (AST) levels, creatine kinase (CK) levels, prolonged activated partial thromboplastin time (aPTT), high viral RNA loads in the serum, and the appearance of neurological manifestations, hemorrhagic features, disseminated intravascular coagulation, and multi-organ failure are also observed during the severe SFTS infection (7-11).”

  1. Line 84: Was the viral RNA extracted from blood? Were blood cells cultured? How can this be quantitative measure of virus copy in the patient?  I am confused about this.

Response: I apologize for the error. Viral RNA was extracted from patients' blood, and virus copy numbers were confirmed using plasmid synthesis and a standard curve. Please refer to the revised manuscript for more details.

  1. Line 105: What are these minor modifications?

Response: I apologize for the error. There were no modifications at all.

  1. Line 124: What is the number of recruited patients? How were they recruited?

Response:  The study involved a total of 470 patients, with 14 hospitals that were IRB approved by each institution conducting the study with suspected SFTS patients who agreed to use clinical records and specimens.

  1. Lines 125-126: If 277 patients were confirmed to have SFTSV infection from 470 patients, how did the authors arrive at 79 patients in the end?

Response: A total of 470 patients with suspected SFTS, who were admitted to hospitals and recruited from the southern part of the Korean peninsula, were enrolled between May 1, 2018, and October 31, 2020. Of these, 277 (58.9%) met the eligibility criteria for laboratory-confirmed SFTSV infection. A total of 81 patients underwent additional blood sampling for the analysis of viral copy numbers and cytokine levels at different time points. Among them, one patient with scrub typhus and SFTS coinfection, and two patients with SFTS-Q fever coinfection, were excluded, leaving 79 patients' specimens for analysis.

  1. Line 129:  Who are the “No specific treatment” -group? Why is the mortality 0% in this group?

Response: Thank you for your comment. In this study, patients who received only conservative treatment such as fluids, without steroid, IVIG, ribavirin, or plasmapheresis, were classified into the no specific treatment group. Since this study is not a randomized controlled trial, it is likely that primarily mild cases received conservative treatment, which may have led to a 0% mortality rate and a high potential for selection bias.

  1. Line 145: About the treatment group.  What other treatments are for the categories of Steroid, Ribavirin, IVIG, and Plasma exchange?

Response: Patients of this study basically received these 5 types treatment only. Among 79 patients, 15 patients two types of treatments together from these 5 types of treatment. Rest of the patients received only 1 treatment of these 5 types treatment. We have also described this point in our manuscript as below:

“15 (18.9%) patients were treated with steroids, 40 (50.6%) patients were treated with immunoglobulin (IVIG), 10 (12.7%) patients were treated with ribavirin, and 14 (17.7%) patients had undergone plasma ex-change (PE). Fifteen patients received more than two specific treatments, and 20 patients did not receive any specific treatment.”

  1. Lines 147-158:  Since the numbers per treatment group as so small, it is difficult to consider statistical analysis relevant.

Response: Thank you for your comment. We agree with your opinion. As your suggestion, we had added following sentence.  “However, due to the small sample sizes in each treatment group, performing meaningful statistical analysis remains challenging.”

  1. Line 209: Please, put the figure headings above the figures.

Response: Thank you for your valuable comment. We have revised figures in our manuscript as per your suggestion.

  1. Line 333: methods were not effective.
  2. Response: We agree with your observation. Since this study is not a randomized controlled trial, we have included the following limitation in our study: 'A limitation of this study is that it was not a randomized controlled trial. Line 353: What is IPA?

Response: The full form of IPA is Invasive pulmonary aspergillosis that is mentioned in the manuscript as below:

“In detail, streptococcal bacteremia, hospital-acquired pneumonia (HAP) due to Pseudomonas aeruginosa and Staphylococcus aureus, urinary tract infection due to Escherichia coli, and invasive pulmonary aspergillosis (IPA) due to Aspergillus fumigatus were shown in the steroid group.”

  1. Line 354: Except for Bae et al. study (remove apostrophe)

Response: Thank you for your valuable comment. We have removed the apostrophe in our revised manuscript as per your suggestion.

Reviewer 2 Report

Comments and Suggestions for Authors

The manuscript describes a retrospective study of the SFTS outbreak. The authors of the manuscript attempted to analyze the success of applied therapeutic strategies used for SFTS therapy in this disease outbreak s 2018-2020. The manuscript does not contain a detailed written introduction. In the introduction chapter detail what treatment strategies are applied for therapy, if the main problem in SFTS is cytokine storm, why pro-inflammatory cytokine blockers are not applied? It is necessary to add information about SFTSV and its transmission agent in the introduction chapter.

The results of the analysis are contradictory and it is not clear what principles were used to form the comparison groups. Table 1 shows that patients without any therapy all survived, while the use of therapy resulted in a high percentage of deaths and ICU admissions. I recommend that the authors reconsider their approach to the formation of comparison groups and expand the materials and methods in the context of describing the selected groups. Describe in which case and patients received therapy for SFTS. It is not clear from the material described whether patients were admitted to the ICU because of the therapeutic strategies used or whether they received therapy after admission to the ICU. An additional recommendation is to compare mono- and combination therapy separately. 

The manuscript has serious flaws in logical composition. Its main conclusions are insufficiently substantiated.

References should be done according of Journal Rules.

Comments on the Quality of English Language Careful editing of the English language is needed.
Minor revisions

Line 103:   «treatmemts»

Line 324:    «a immunoregulatory»

Line 333:    «methods was not…»

Line 385:    «inflamatory»

Author Response

Reviewer 2

Comments and Suggestions for Authors

  1. The manuscript describes a retrospective study of the SFTS outbreak. The authors of the manuscript attempted to analyze the success of applied therapeutic strategies used for SFTS therapy in this disease outbreak s 2018-2020. The manuscript does not contain a detailed written introduction. In the introduction chapter detail what treatment strategies are applied for therapy, if the main problem in SFTS is cytokine storm, why pro-inflammatory cytokine blockers are not applied? It is necessary to add information about SFTSV and its transmission agent in the introduction chapter.

Response: Thank you for your valuable comment. We have revised our manuscript as per your suggestion and rewrote the line as below:

“Severe fever with thrombocytopenia syndrome (SFTS) is an acute febrile illness characterized by fever, thrombocytopenia, and leukopenia. It is caused by the SFTS virus (SFTSV), which was first identified in China in 2011 (1). SFTSV belongs to the Phlebovirus genus, a member of the Bunyaviridae family (2). This syndrome is a tick-borne zoonosis, meaning the virus is transmitted to humans through tick bites. The infection spreads from animals to humans and can also be transmitted from human to human (3).

Although it has been 10 years since SFTS was first recognized, we still treat it conservatively due to the lack of a suitable treatment option; as a result, the mortality rate remains high, approximately 6% in China and 20% in Korea (4, 5). Several specific treatments, such as ribavirin, plasma exchange, steroids, tocilizumab, and intravenous immunoglobulin, are administered to patients with severe SFTS. A previous study found that SFTS patients who received plasma exchange treatment early in the onset of symptoms survived for a longer duration than those in the non-plasma exchange group (6). However, they did not compare cytokine levels between the two groups, and there is limited data on cytokine concentrations in the plasma exchange versus non-plasma exchange groups.

The cytokine storm plays a vital role in the pathogenesis of SFTS. Steroid treatment is considered a therapeutic option to suppress this cytokine storm in severe SFTS patients (7). Recently, tocilizumab has also been used to mitigate the cytokine storm in SFTS patients. Some studies indicate that tocilizumab has significant clinical advantages when used in combination with corticosteroids (8, 9). Regarding steroid treatment, our previous study and one conducted in Japan found that steroids can increase complications among SFTSV-positive patients (10, 11). Additionally, we observed that patients in the steroid group (Acute Physiology and Chronic Health Evaluation II [APACHE II] score <14) with mild conditions showed significantly shorter survival times compared to those in the non-steroid group (10). Therefore, therapy associated with cytokine storm modification should be approached with caution.”

The reviewer also inquired why we did not use pro-inflammatory cytokine blockers to reduce the cytokine storm. The main purpose of this study was to verify whether various specific treatments for SFTSV could reduce cytokine levels among infected patients. Our findings indicated that the different treatments used to address SFTSV were unable to reduce cytokines. Therefore, further well-controlled studies are needed to explore the treatment of SFTS-infected patients with cytokine blockers, such as tocilizumab, alongside specific treatments.

  1. The results of the analysis are contradictory and it is not clear what principles were used to form the comparison groups. Table 1 shows that patients without any therapy all survived, while the use of therapy resulted in a high percentage of deaths and ICU admissions. I recommend that the authors reconsider their approach to the formation of comparison groups and expand the materials and methods in the context of describing the selected groups. Describe in which case and patients received therapy for SFTS. It is not clear from the material described whether patients were admitted to the ICU because of the therapeutic strategies used or whether they received therapy after admission to the ICU. An additional recommendation is to compare mono- and combination therapy separately. 

Response: Thank you for your comment. In this study, patients who received only conservative treatment such as fluids, without steroid, IVIG, ribavirin, or plasmapheresis, were classified into the no specific treatment group. Of the patients enrolled in this study, all patients treated in the ICU were not admitted to the ICU because of the specific treatment strategy used to analyze the treatment effect and were treated in the ICU from the first day of hospitalization because the severity of SFTS disease itself was high at the time of admission. Since this study is not a randomized controlled trial, it is likely that primarily mild cases received conservative treatment, which may have led to a 0% mortality rate and a high potential for selection bias.

  1. The manuscript has serious flaws in logical composition. Its main conclusions are insufficiently substantiated.

Response: Thank you for your comment. This is not a randomized controlled trial but an observational study. However, there are no existing articles that address the comparison of viral copy numbers and cytokines before and after various specific treatments for SFTSV. The main focus is not on comparing efficacy between different treatment groups but on assessing the treatment's efficacy by evaluating changes before and after the specific treatment within an observational framework.

  1. References should be done according of Journal Rules.

Response: Thank you for your valuable comment. We have revised references of our manuscript according to Journal’s rule.

Reviewer 3 Report

Comments and Suggestions for Authors

Analysis of changes in viral load and inflammatory cytokines, as well as the occurrence of secondary infections, in SFTS patients treated with specific treatments: a prospective multicenter cohort study” by Jun-Won Seo et al. describe studies that a scientific evidence-based treatment can improve prognosis through changes in viral load and inflammatory cytokines according to the specific treatment of SFTS patients. Specific comments have been included below.

1. According to previous reports, the incidence of SFTS is higher in males than in females. The proportions presented in the article reflect hospitalization rates, and it is unclear whether they accurately reflect the actual incidence rates.

2. SFTSV can be divided into Chinese and Japanese clades, and there are differences in mortality between different clades. The author mentioned measuring the viral load of SFTSV, but it is unclear whether sequencing analysis was performed to determine the specific strain of virus infection. Further research is needed to determine if there are differences in treatment approaches for infections with different virus strains.

3. The authors analyzed four specific treatment methods, and the results showed that their efficacy were not superior to that of the untreated group. This situation is confusing because the untreated group had a 100% survival rate, while the treated group may have an increased risk of mortality. The significance of using these drugs in the current treatment stage still requires further research and validation.

4. Apart from the discussion section, there is no conclusive text in the article. It simply provides explanations for the information presented in the figures and tables.

Author Response

Reviewer 3

  1. According to previous reports, the incidence of SFTS is higher in males than in females. The proportions presented in the article reflect hospitalization rates, and it is unclear whether they accurately reflect the actual incidence rates.

Response: Thank you for your comment.

Incidence rate of SFTSV between male and female can be varied from country to country. Study of Choi et al. revealed the incidence rate of SFTSV at South Korea was 50.8% in male, and 49.2 % in female. On the other hand, study of Hidaka et al. presented that the incidence rate of SFTSV was more among female (53%) than male (47%) in Japan. Furthermore, the study of Guo et al. presented that the incidence of SFTSV between female and male was same in China. Therefore, we can say that this incidence of SFTSV is not always higher in males than females. It doesn't seem like the differences between men and women are that significant. Rather than, the incidence of SFTSV is varied from country to country. In addition, this study focuses on hospitalized patients only. Therefore, it does not include those with mild symptoms who were not admitted. As a result, the actual incidence rate is difficult to ascertain.

We have attached below a reference below for better understanding the equal positivity rate among SFTS patients.

  • “Guo, C-T., Q-B. Lu, S-J. Ding, C-Y. Hu, J-G. Hu, Y. Wo, Y-D. Fan et al. "Epidemiological and clinical characteristics of severe fever with thrombocytopenia syndrome (SFTS) in China: an integrated data analysis." Epidemiology & Infection144, no. 6 (2016): 1345-1354.”
  • Hidaka, Kazuhiro, Shuya Mitoma, Junzo Norimine, Masayuki Shimojima, Yoshiki Kuroda, and Takuji Hinoura. "Seroprevalence for severe fever with thrombocytopenia syndrome virus among the residents of Miyazaki, Japan: An epidemiological study." Journal of Infection and Chemotherapy 30, no. 6 (2024): 481-487.
  • Choi, Seong Jin, Sang-Won Park, In-Gyu Bae, Sung-Han Kim, Seong Yeol Ryu, Hyun Ah Kim, Hee-Chang Jang et al. "Severe fever with thrombocytopenia syndrome in South Korea, 2013-2015." PLoS neglected tropical diseases 10, no. 12 (2016): e0005264.
  1. SFTSV can be divided into Chinese and Japanese clades, and there are differences in mortality between different clades. The author mentioned measuring the viral load of SFTSV, but it is unclear whether sequencing analysis was performed to determine the specific strain of virus infection. Further research is needed to determine if there are differences in treatment approaches for infections with different virus strains.

Response: Thank you for your comment. We only checked the positivity rate among patients with different treatments by real time pcr, and checked the viral load. In addition, we have also checked the serum cytokines changes. However, we did not identify different clades of SFTS through genomic sequencing. That is why we have mentioned in our study that we need to do further study. However, we have revised our manuscript as below by considering your comment:

“Further research on this issue is required, with more patients with SFTS in a prospective cohort to perform the genomic sequences and find out if there is any symptom difference among SFTS positive patients.”  

  1. The authors analyzed four specific treatment methods, and the results showed that their efficacy were not superior to that of the untreated group. This situation is confusing because the untreated group had a 100% survival rate, while the treated group may have an increased risk of mortality. The significance of using these drugs in the current treatment stage still requires further research and validation.

Response: Thank you for your comment. In this study, patients who received only conservative treatment such as fluids, without steroid, IVIG, ribavirin, or plasmapheresis, were classified into the no specific treatment group. Since this study is not a randomized controlled trial, it is likely that primarily mild cases received conservative treatment, which may have led to a 0% mortality rate and a high potential for selection bias. As you suggested, the significance of using these drugs at the current stage of treatment still requires further research and validation.

  1. Apart from the discussion section, there is no conclusive text in the article. It simply provides explanations for the information presented in the figures and tables.

Response: Thank you for your comment. While this manuscript presents experimental data, the primary focus is to demonstrate that plasmapheresis and steroid treatment do not effectively reduce cytokine levels or viral load. In South Korea, these treatments are commonly used to manage cytokine storm. Our study yields result that contrast with the expected outcomes of these treatments, based on an observational assessment of changes before and after the specific treatments. We recommend conducting a randomized controlled trial to further investigate the differences in cytokine levels and viral load before and after these treatments.

Round 2

Reviewer 1 Report

Comments and Suggestions for Authors

I agree with the revisions. 

Reviewer 2 Report

Comments and Suggestions for Authors

Recommended for publication

Reviewer 3 Report

Comments and Suggestions for Authors

None